# Do the associations of body mass index and waist circumference with back pain change as people age? 32 years of follow-up in a British birth cohort

Stella Muthuri,[1] Rachel Cooper [ID],[2] Diana Kuh [ID],[1] Rebecca Hardy[3]

[1]MRC Unit for Lifelong Health and Ageing, UCL, London, UK
[2]Department of Sport and Exercise Sciences, Musculoskeletal Science and Sports Medicine Research Centre, Manchester Metropolitan University, Manchester, UK
[3]CLOSER, Social Research Institute, UCL, London, UK

**Correspondence to**
Rebecca Hardy;
rebecca.hardy@ucl.ac.uk

## ABSTRACT

**Objectives** To investigate whether cross-sectional and longitudinal associations of body mass index (BMI) and waist circumference (WC) with back pain change with age and extend into later life.

**Design** British birth cohort study.

**Setting** England, Scotland and Wales.

**Participants** Up to 3426 men and women from the MRC National Survey of Health and Development.

**Primary outcome measures** Back pain (sciatica, lumbago or recurring/severe backache all or most of the time) was self-reported during nurse interviews at ages 36, 43, 53 and 60–64 years and in a postal questionnaire using a body manikin at age 68.

**Results** Findings from mixed-effects logistic regression models indicated that higher BMI was consistently associated with increased odds of back pain across adulthood. Sex-adjusted ORs of back pain per 1 SD increase in BMI were: 1.13 (95% CI: 1.01 to 1.26), 1.11 (95% CI: 1.00 to 1.23), 1.17 (95% CI: 1.05 to 1.30), 1.31 (95% CI: 1.15 to 1.48) and 1.08 (95% CI: 0.95 to 1.24) at ages 36, 43, 53, 60–64 and 68–69, respectively. Similar patterns of associations were observed for WC. These associations were maintained when potential confounders, including education, occupational class, height, cigarette smoking status, physical activity and symptoms of anxiety and depression were accounted for. BMI showed stronger associations than WC in models including both measures.

**Conclusions** These findings demonstrate that higher BMI is a persistent risk factor for back pain across adulthood. This highlights the potential lifelong consequences on back pain of the rising prevalence of obesity within the population.

## Strengths and limitations of this study

► The availability of data on back pain, body mass index (BMI) and waist circumference (WC) prospectively ascertained over 32 years of follow-up in a large representative population-based sample is a key and unique strength of this study.

► This allowed us to formally test whether the cross-sectional and longitudinal relationships of BMI and WC with back pain change with age, extend into later life and are independent of each other so addressing important gaps in our current understanding.

► As back pain was not assessed in exactly the same way at different time points across adulthood, timing of onset of pain was not recorded and severity and chronicity were not distinguished, results need to be interpreted with some caution.

► Residual confounding cannot be fully ruled out as a potential explanation of our findings however, the risk of this has been minimised by taking into account of a wide range of potential confounders measured prospectively across adulthood.

► Cross-sectional associations between body size and back pain may be explained by reverse causality, but we also investigated longitudinal associations and our conclusions remained unchanged.

public health challenge of the burden of low back pain.[9]

Obesity, usually indicated by high body mass index (BMI), has been identified as an important risk factor for back pain (typically self-reported during a structured interview or self-completion questionnaire).[10–14] However, the majority of existing studies have examined cross-sectional associations at a single time-point, often in early or mid-adulthood[10 13–19] and/or are in occupational-based cohorts.[10] There are few studies on obesity and back pain in older populations. This is despite evidence to suggest that the aetiology of back pain may change with age as degenerative back disorders become increasingly common; these disorders may

## INTRODUCTION

Back pain is one of the most commonly reported musculoskeletal disorders and was recently ranked as the number one cause of disability in most countries worldwide.[1] As it is a major cause of activity limitation and functional decline in later life,[2–8] the fact that its prevalence is projected to increase as population ageing continues is a considerable cause for concern. As a result, there have been calls to intensify research efforts to address the

have different risk factors to conditions that precipitate back pain earlier in life. This is coupled with reduced pain-modulatory capacity occurring with advancing age partly explaining increases in persistent and disabling pain conditions among older adults.[20] In one study of older populations (age >50 years) from nine countries, high BMI was associated with increased odds of back pain, assessed during an interview using a simple question on experience of back pain in the last 30 days, in European countries (Poland, Russia, Finland and Spain) and South Africa.[21] However, whether or not the associations found among these older adults were weaker, due to potential changes in the underlying aetiology of back pain, or of similar strength to those that would have been observed in these populations if they had been assessed at younger ages could not be established. To the best of our knowledge, no study has formally investigated whether the association between BMI and back pain changes with age; thus whether or not obesity remains a suitable target for intervention for the prevention of back pain into later life, alongside the important role of interventions targeting obesity to prevent many other chronic conditions, remains to be established.

Higher central adiposity, as well as higher BMI, has been related to increased risk of back pain[12 15–19 22] and may be more indicative of inflammatory processes due to increases in fat mass. Despite this, few studies[22] have compared the strength of associations between BMI and abdominal adiposity or tested whether there is any additional effect of abdominal adiposity over and above BMI. This could provide new aetiological insights to help inform the development of interventions to prevent or alleviate back pain.

The MRC National Survey of Health and Development (NSHD), the oldest of the British birth cohort studies, has assessed BMI, waist circumference (WC) and back pain at multiple time points across adulthood from mid-life up to age 69. We therefore address important research gaps by first examining whether the cross-sectional relationships of BMI and WC with back pain change with age, extend into later life and are independent of each other. We then assess the prospective associations of BMI and WC with back pain at the subsequent age to assess the extent to which patterns of association with age may be affected by reverse causality.

## SUBJECTS AND METHODS
The NSHD is a socially stratified sample of 5362 single, legitimate births that occurred in England, Wales and Scotland in 1 week of March 1946. To date, the same participants have been assessed on up to 24 occasions between birth and age 69, with participation rates remaining relatively high (ie, >80% of eligible sample responding) across life.[23–25]

### Patient and public involvement
Over the 74 years of this study, the research has increasingly involved participants, in line with changing norms about conducting cohort studies. Participant involvement includes receiving personal letters from the research team as required, and invitations to participate in birthday celebrations, public engagement activities and focus groups to discuss future data collections. When piloting new questionnaires and assessments, including the most recent of those used in these analyses, patients from general practices and the University College London Hospitals Patient Public Involvement group were recruited and asked to provide feedback which was taken into account when designing the mainstage fieldwork.

### Back pain assessment
Back pain was ascertained during five of the main assessments of NSHD participants via structured nurse interviews at ages 36, 43, 53 and 60–64 and in a postal questionnaire at age 68 (see Supplementary Methods for questions used). At all ages except age 68, participants were asked whether they had sciatica, lumbago or recurring/severe backache all or most of the time (ever at ages 36 and 43 and in the previous 12 months at ages 53 and 60–64). At age 68, participants were asked whether they had experienced any ache or pain in the previous month which had lasted for 1 day or longer, not including pain occurring during the course of a feverish illness such as influenza. Those who responded positively were asked to shade the location of their pain using a four-view body manikin. Those who shaded any back site were classified as having back pain for the purposes of these analyses.

### Anthropometric measurements
Height, weight and WC (taken at the midpoint between the costal margin and iliac crest) were measured by nurses using standardised protocols at ages 36, 43, 53, 60–64 and 69 years. BMI was calculated at each age as weight (kg)/height (m$^2$). BMI and WC at each age were sex-standardised (to a mean of 0 and SD of 1 (calculated as $(x\text{-mean}_i)/\text{SD}_j$ where $x$ is the raw measure and mean$j$ and SD$j$ are the sample mean and SD for sex $j$); our units of analysis for BMI and WC are both 1 SD to facilitate comparisons of effect sizes across age and sex.

### Fat and lean mass at 60–64 years
Measures of body composition were obtained in participants who attended one of six Clinical Research facilities when aged 60–64 years using a QDR 4500 Discovery DXA scanner (Hologic, Bedford, MA, USA) while the individuals were in a supine position. Whole body fat mass was measured and whole body lean mass was calculated as total mass minus fat and bone mass. Mass from the head was excluded and measures were converted to kilogram. Full details are provided elsewhere.[26] Fat mass index (kg/m$^2$) and lean mass index (kg/m$^2$) were calculated by dividing the measures of mass by height-squared.

### Covariates
Covariates representing different domains of the biopsychosocial model of pain were selected a priori.[27] This included variables that have previously been identified

as key risk factors for back pain and could potentially confound the main associations.[28–30] These were sex, educational attainment at age 26, occupational class at age 53 and measures of height, smoking status, physical activity, and symptoms of anxiety and depression, which were available at ages 36, 43, 53, 60–64 and 68–69 and so included as time-varying covariates.

The highest education level achieved by age 26 was grouped into no qualifications, up to O-level or equivalent, or A-level or equivalent and above. Own occupation at age 53 was categorised according to the Registrar General's social classification into three groups: high (I or II: professional, managerial or technical); middle (IIINM skilled non-manual or IIIM: skilled manual); low (IV or V: partly skilled or unskilled manual).

Smoking status was assessed by self-report at each wave and categorised as never, ex and current smoker. Participation in sports, vigorous leisure activities or exercise was assessed at each wave and participants were grouped as inactive, moderately active (1–4 times/month) or regularly active (≥5 times/month).[31]

Symptoms of anxiety and depression were also assessed at all five waves. At age 36, a shortened version of the Present State Examination[32] was used at a nurse interview and the total score was recoded to a binary variable using the recommended threshold for caseness of 5 or more. At age 43, participants completed the 18-item Psychiatric Symptom Frequency (PSF) scale[33] and the total score was calculated and then dichotomised into the absence of symptoms (PSF score <23) and the presence of symptoms (PSF score ≥23). At ages 53, 60–64 and 68-69 the 28-item General Health Questionnaire (GHQ-28)[34] was self-administered and scores for all items were summed and further dichotomised using a recommended threshold for caseness of 5 or more.

## Statistical analysis

We first examined descriptive statistics for each variable and formally tested sex differences in their distributions using $\chi^2$ and t-tests as appropriate.

Multilevel mixed-effects logistic regression models with a random intercept were then used to test associations between BMI and back pain with measurement occasion nested within individuals. These models allow for the correlation between measurements on the same individual. We fitted measurement wave as a categorical variable with age 36 as the reference category and adjusted for sex. Interactions between wave and sex were tested in the model however, as there was no evidence of interactions, these were removed in subsequent models. We then added BMI as a time-varying covariate. Formal assessment of whether associations between BMI and back pain varied by sex were performed by including sex by BMI interaction terms in models and where no evidence of interaction was found models were sex-adjusted. Deviations from linearity were assessed by including quadratic terms for BMI, but no evidence of this was found. To assess whether the association of BMI with back pain changed with

age, interactions between wave and BMI were included (model 1). The model was then adjusted for potential confounding variables (model 2). Analyses were repeated replacing BMI with WC and then a final confounder adjusted model was run in which BMI and WC were included together. All models included the maximum number of participants, which was those participants with a valid measure of back pain, BMI, WC and all covariates for at least one age (n=3426) (see online supplemental table S1 for number of participants who contributed data at each wave).

We then repeated the same modelling approach to assess prospective associations between BMI, WC and back pain. To achieve this, we used back pain at 43, 53, 60–64 and 68 as outcomes and related them to BMI and WC at the prior age (ie, at ages 36, 43, 53 and 60–64). These models included a sample size of 3044.

Finally, in the subsample with body composition measures at age 60–64 (n=1186) supplementary analyses using multiple regression models, assessed the associations between fat mass index and lean mass index at 60–64 years and back pain at 68 years. Model 1 included sex and both body composition measures and model 2 added the same covariates used in previous models.

All analyses were performed using STATA V.14.1.

## RESULTS

The characteristics of the study sample are shown in table 1 (and online supplemental table S2). Overall, the percentage reporting back pain varied from 16.7% to 34.0% and was higher in women than men at every age (table 1).

There was no evidence of an interaction between sex and either BMI (p=0.2) or WC (p=0.9) and so the main models were sex-adjusted. In sex-adjusted models, higher BMI was associated with increased odds of back pain at every age (table 2). Sex-adjusted ORs of back pain per 1 SD increase in BMI estimated from this model were 1.13 (95% CI: 1.01 to 1.26), 1.11 (95% CI: 1.00 to 1.23), 1.17 (95% CI: 1.05 to 1.30), 1.31 (95% CI: 1.15 to 1.48) and 1.08 (95% CI :0.95 to 1.24) at ages 36, 43, 53, 60–64 and 68–69, respectively. There was no evidence of a difference in association between ages 36 and 43, 53 or 68 (wave by BMI interactions p>0.5) and only weak evidence that the association at ages 60–64 was stronger than at age 36 (ages 60–64 wave by BMI interaction: p=0.07) (online supplemental table S3). Adjustment for potential confounders had minimal impact on these findings and estimated ORs for each age remained similar (table 2).

A similar, and slightly less variable, pattern of association was observed for WC, with no evidence that the association at age 60–64 was stronger than at other ages (wave by BMI interaction: p=0.2) (table 2, online supplemental table S3). Adjustment for potential confounders attenuated all ORs somewhat, and to a greater extent than the ORs for BMI (table 2).

**Table 1** Characteristics of the MRC National Survey of Health and Development sample included in analysis*

| | Male | Female | P value† |
|---|---|---|---|
| **Back pain, N (%), at age** | | | |
| 36 years | | | 0.08 |
| No | 1283 (83.3) | 1256 (80.9) | |
| Yes | 257 (16.7) | 297 (19.1) | |
| 43 years | | | 0.06 |
| No | 1106 (74.6) | 1042 (71.5) | |
| Yes | 377 (25.4) | 415 (28.6) | |
| 53 years | | | 0.06 |
| No | 925 (69.4) | 903 (66.0) | |
| Yes | 407 (30.6) | 465 (34.0) | |
| 60–64 years | | | 0.06 |
| No | 629 (72.1) | 643 (68.1) | |
| Yes | 243 (27.9) | 301 (31.9) | |
| 68 years | | | 0.02 |
| No | 637 (74.1) | 635 (69.2) | |
| Yes | 223 (25.9) | 282 (30.8) | |
| **BMI (kg/m², mean (SD), at age** | | | |
| 36 years | 24.8 (3.3) | 23.5 (4.1) | <0.001 |
| 43 years | 25.7 (3.5) | 25.1 (4.7) | <0.001 |
| 53 years | 27.4 (4.1) | 27.4 (5.4) | 0.85 |
| 60–64 years | 27.9 (4.1) | 27.9 (5.5) | 0.99 |
| 69 years | 28.1 (4.5) | 28.0 (5.6) | 0.93 |
| **Waist circumference (cm), mean (SD), at age** | | | |
| 36 years | 89.5 (9.3) | 76.9 (11.5) | <0.001 |
| 43 years | 91.8 (9.9) | 77.7 (11.2) | <0.001 |
| 53 years | 97.7 (10.9) | 85.8 (12.8) | <0.001 |
| 60–64 years | 100.8 (11.1) | 92.2 (13.0) | <0.001 |
| 69 years | 100.9 (12.3) | 92.1 (14.0) | <0.001 |

*Maximum n=3426 (1723 males and 1703 females) (this includes participants with a valid measure of back pain, BMI, waist circumference and each covariate for at least one age). Number of participants contributing data at each age: 36 years, n=3093; 43 years, n=2940; 53 years, n=2700; 60–64 years, n=1816; 68–69 years, n=1777 (see online supplemental table S1) for more details.
†P value from formal tests of sex differences using $\chi^2$ and t-tests as appropriate.
BMI, body mass index; WC, waist circumference.

Including both BMI and WC in the same fully adjusted model with all age interaction terms led to inflated standard errors. We therefore included only the age 60–64 wave by BMI and the age 60–64 wave by WC interaction, finding stronger evidence of the interaction with BMI. Our final fully adjusted model thus included only the interaction between BMI and age 60–64; the association of BMI with back pain (OR per SD (95% CI): 1.11 (0.99 to 1.24) at 36, 43, 53 and 68–69 years and 1.28 (1.09 to 1.50) at 60–64 years) was stronger than that for WC (OR per SD (95% CI): 1.02 (0.91 to 1.13)) at all ages.

Findings for both BMI and WC were very similar in the longitudinal analysis, when relating the body size measure

at the prior age to subsequent back pain (table 3). In supplementary analyses (n=1186), higher fat mass index was associated with higher odds of back pain at age 68 (OR per SD (95% CI): 1.23 (1.04 to 1.45)) in a model adjusting for sex and lean mass index, but no association was observed for lean mass index (online supplemental table S4). The association with fat mass index was maintained after adjustment for other potential confounders.

## DISCUSSION

In a large nationally representative British birth cohort, higher BMI and WC were consistently associated with increased odds of back pain between ages 36 and 68. These associations were maintained after adjustment for confounders and, BMI remained more strongly related to back pain than WC in a model including both variables. Findings were similar when prospective associations were investigated for back pain between 43 and 68.

Our findings are consistent with previous studies which have reported associations between overweight and obesity, indicated by high BMI, and increased odds of back pain.[10] Findings from cross-sectional studies also suggest positive associations between higher WC and low back pain in women, but not in men.[12 15–19] These previous studies have sampled participants across a range of different ages (15 years and older) and although they have adjusted for age, none has examined whether associations of BMI and WC with back pain vary with age or whether associations remain into older age. Our study shows persisting associations of BMI and WC with back pain until age 68 years with little evidence of variation with age.

Another key finding is that BMI exhibited a stronger association than WC in a model including both measures. This suggests that mechanisms specifically related to central adiposity do not fully underlie the associations observed. This contrasts with a previous study in young adults (age 24–39 years) which found that WC but not BMI remained associated with back pain in mutually adjusted models in women only.[15] Our findings in relation to fat and lean mass suggest that associations with BMI may be driven by whole body fat mass, rather than abdominal fat mass specifically. In testing these associations, we are addressing the call made in a recent systematic review for further high-quality studies on the relationships of body composition with pain.[35] Our findings are consistent with a prospective study[12] included in this review that found evidence that fat mass was associated with low back pain intensity and disability in both sexes. Likewise, a cross-sectional study found that higher fat mass (independent of lean mass) was associated with higher levels of low back pain intensity and disability but found no relationship with lean mass after adjusting for fat mass.[11] The stronger associations with BMI than WC may point to the importance of mechanical effects, with increased BMI resulting in higher compressive force on the spine during activity.[10] However, we cannot rule out

**Table 2** ORs of back pain at each age per 1 SD increases in BMI and waist circumference at the same age estimated from multilevel logistic models (12 326 observations nested within 3426 individuals)

| | Model 1* | | Model 2† | |
| --- | --- | --- | --- | --- |
| | OR (95% CI) | P value | OR (95% CI) | P value |
| **BMI** | | | | |
| **Per SD BMI at** | | | | |
| 36 years | 1.13 (1.01 to 1.26) | 0.03 | 1.12 (1.00 to 1.25) | 0.05 |
| 43 years | 1.11 (1.00 to 1.23) | 0.06 | 1.11 (1.00 to 1.23) | 0.05 |
| 53 years | 1.17 (1.05 to 1.30) | 0.003 | 1.17 (1.05 to 1.30) | 0.003 |
| 60–64 years | 1.31 (1.15 to 1.48) | <0.001 | 1.30 (1.14 to 1.47) | <0.001 |
| 68 years | 1.08 (0.95 to 1.24) | 0.2 | 1.07 (0.94 to 1.22) | 0.3 |
| **Waist circumference** | | | | |
| **Per SD WC at** | | | | |
| 36 years | 1.17 (1.05 to 1.31) | 0.005 | 1.14 (1.02 to 1.27) | 0.02 |
| 43 years | 1.12 (1.01 to 1.24) | 0.03 | 1.08 (0.98 to 1.20) | 0.1 |
| 53 years | 1.16 (1.05 to 1.29) | 0.005 | 1.13 (1.02 to 1.25) | 0.03 |
| 60–64 years | 1.27 (1.12 to 1.44) | <0.001 | 1.21 (1.07 to 1.38) | 0.003 |
| 68 years | 1.12 (0.98 to 1.27) | 0.1 | 1.07 (0.94 to 1.22) | 0.3 |

Please see online supplemental table S5 for results from analyses rerun with BMI and WC modelled in raw units, that is, $kg/m^2$ and cm, respectively.
*Model 1: Includes age as a categorical variable, standardised BMI or WC (mean=0, SD=1) and an age by BMI or age by WC interaction (as appropriate), adjusted for sex (as there was no evidence of a sex by BMI (p=0.2) or a sex by WC (p=0.9) interaction).
†Model 2: Includes age as a categorical variable, standardised BMI or WC (mean=0, SD=1) and an age by BMI or age by WC interaction (as appropriate), adjusted for sex, education, occupational class and time-varying covariates (height, cigarette smoking status, physical activity and symptoms of anxiety and depression).
BMI, body mass index; WC, waist circumference.

**Table 3** ORs of back pain at each age per 1 SD increase in BMI and waist circumference at the previous age estimated from multilevel logistic models (8595 observations nested within 3044 individuals)

| | | Model 1* | | Model 2† | |
| --- | --- | --- | --- | --- | --- |
| **Per SD BMI/WC at age** | **Back pain at age** | OR (95% CI) | P value | OR (95% CI) | P value |
| **BMI** | | | | | |
| 36 years | 43 years | 1.08 (0.97 to 1.20) | 0.2 | 1.06 (0.96 to 1.18) | 0.3 |
| 43 years | 53 years | 1.17 (1.05 to 1.31) | 0.004 | 1.16 (1.04 to 1.29) | 0.009 |
| 53 years | 60–64 years | 1.32 (1.15 to 1.51) | <0.001 | 1.29 (1.13 to 1.48) | <0.001 |
| 60–64 years | 68 years | 1.11 (0.96 to 1.28) | 0.2 | 1.08 (0.94 to 1.25) | 0.3 |
| **Waist circumference** | | | | | |
| 36 years | 43 years | 1.12 (1.01 to 1.24) | 0.04 | 1.08 (0.97 to 1.21) | 0.1 |
| 43 years | 53 years | 1.17 (1.05 to 1.30) | 0.004 | 1.14 (1.02 to 1.26) | 0.02 |
| 53 years | 60–64 years | 1.28 (1.12 to 1.47) | <0.001 | 1.23 (1.07 to 1.40) | 0.003 |
| 60–64 years | 68 years | 1.10 (0.96 to 1.26) | 0.2 | 1.06 (0.92 to 1.21) | 0.4 |

Please see online supplemental table S6 for results from analyses rerun with BMI and WC modelled in raw units, that is, $kg/m^2$ and cm, respectively.
*Model 1: Includes age as a categorical variable, standardised BMI or WC (mean=0, SD=1) and an age by BMI or age by WC interaction (as appropriate), adjusted for sex (as there was no evidence of a sex by BMI (p=0.5) or a sex by WC (p=0.9) interaction).
†Model 2: Includes age as a categorical variable, standardised BMI or WC (mean=0, SD=1) and an age by BMI or age by WC interaction (as appropriate), adjusted for sex, education, occupational class and time-varying covariates (height, cigarette smoking status, physical activity and symptoms of anxiety and depression).
BMI, body mass index; WC, waist circumference.

the fact that chronic inflammatory or metabolic pathways may also be relevant. For example, atherosclerosis and chronic inflammation, which are linked to obesity, have also been associated with disc degeneration and low back pain.[36 37]

A key strength of our study is the large representative population-based sample of adults with assessments of back pain and measured height, weight and WC at five time points over 32 years of follow-up. We assessed back pain using simple questions asked during structured nurse interviews and in a self-completed questionnaire which are commonly used methods in population-based studies of back pain.[10–14 21] The prevalence estimates of back pain in our study at different ages were comparable to those reported in other studies,[2] and we were able to show that associations with BMI are consistent across adulthood and persist into old age. Another strength is the availability of data on a wide range of potential confounders measured prospectively across adulthood that allowed us to adjust for a range of time-varying covariates and reduce the risk of residual confounding.

Alongside key strengths, our results need interpreting in the context of a number of potential limitations. First, while our study population were selected to be nationally representative at birth and due to high participation rates across life have remained so in many respects,[23–25] bias may have been introduced due to the necessary exclusion of all those lost to follow-up before age 36 and/or with missing data on back pain, body size and covariates at all five waves. Second, cross-sectional associations observed between body size and back pain may be at least partly explained by reverse causality, especially at the latest wave when back pain was measured at age 68 years, a year prior to the assessment of body size due to the design of the data collection at age 68–69 years. However, we also investigated associations between back pain and BMI measured at a prior age and findings remained unchanged. A third potential limitation, relating to the fact that our analyses are posthoc, that is, data are drawn from a large population-based study designed to capture information on a wide range of different measures of health and their risk factors across life rather than to address this specific research question, is that back pain was not assessed in exactly the same way at different time points across adulthood. In addition, it was not possible to establish the severity and chronicity of back pain, and the validity of the back pain assessments used has not been evaluated. Finally, our study population comprised Caucasian men and women born in Britain in 1946. While this allows us to rule out confounding by age or ethnicity as potential explanations of our findings, it may limit their generalisability. Further research is thus necessary to consider the implications of these findings for more ethnically diverse cohorts and also more recently born cohorts, the latter who are likely to have spent more of their lives overweight or obese.[38]

In summary, our findings suggest that there is a consistent relationship between higher adiposity and back pain across adulthood. Our results also show that WC does not add further information once BMI has been accounted for in this relationship. This underscores the importance of primary and secondary interventions to prevent excessive weight gain and reduce overweight and obesity across the whole of adulthood in order to help prevent back pain and its disabling consequences as individuals age.

**Acknowledgements** The authors are grateful to NSHD study members for their continuing participation in the study. We also thank members of the NSHD scientific and data collection teams.

**Contributors** SM and RH had full access to all the data in the study and take full responsibility for the integrity and the accuracy of data analysis. SM, DK and RC conceived the idea for this study; SM, DK, RH and RC contributed to the development of the study objectives; DK, RH and RC acquired the data; SGM and RH analysed the data; SM and RH drafted the manuscript; all authors contributed to the manuscript's critical revision and provided final approval of the version to be published.

**Funding** This work was supported by the UK Medical Research Council (Programme codes: MC_UU_12019/4 and MC_UU_12019/2). RH is Director of the CLOSER consortium which is supported by funding from the Economic and Social Research Council (ESRC) (award reference: ES/K000357/1).

**Competing interests** None declared.

**Patient consent for publication** Not required.

**Ethics approval** All waves of data collection have complied with ethical standards. Ethical approval for the most recent data collection at age 68–69 was obtained from the Queen Square Research Ethics Committee (14/LO/1073) and the Scotland A Research Ethics Committee (14/SS/1009). All methods were carried out in accordance with the relevant guidelines and regulations and written informed consent was obtained.

**Provenance and peer review** Not commissioned; externally peer reviewed.

**Data availability statement** Data may be obtained from a third party and are not publicly available. Data used in this publication are available to bona fide researchers upon request to the NSHD Data Sharing Committee via a standard application procedure. Further details can be found at http://www.nshd.mrc.ac.uk/data. doi: 10.5522/NSHD/Q101; doi: 10.5522/NSHD/Q102; 10.5522/NSHD/Q103.

**ORCID iDs**
Rachel Cooper http://orcid.org/0000-0003-3370-5720
Diana Kuh http://orcid.org/0000-0001-7386-2857

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
