## [Reviewer comments · BMJ Open]

ARTICLE DETAILS

TITLE (PROVISIONAL)	Do the associations of body mass index and waist circumference with back pain change as people age? 32 years of follow-up in a British birth cohort
AUTHORS	Muthuri, Stella; Cooper, Rachel; Kuh, Diana; Hardy, Rebecca

VERSION 1 – REVIEW

REVIEWER	Ingrid Heuch Department of Research, Innovation and Education, Division of Clinical Neuroscience, Oslo University Hospital, Oslo, Norway
REVIEW RETURNED	12-May-2020

GENERAL COMMENTS	The study investigates associations of body mass index (BMI) and waist circumference with back pain. The paper is well-written and includes data collected over a long period of time. This represents a valuable source of data compared to other studies of back pain. I would have preferred that the authors had included as references more papers of studies of the association between these anthropometric measures and back pain, and I have suggested the three studies published in papers mentioned below. Especially I think it is important that the authors are aware of the third paper I mention, which deals with a comparison of anthropometric measures and back pain in a follow-up setting. The first paper I recommend that the authors should be aware of included 30 102 men and 33 866 women with information on BMI and chronic low back pain. The age-groups included were 20-39 years, 40-49 years, 60-79 years, and the last age group included participants who were at least 80 years old. Relations were assessed by logistic regression of low back pain with respect to BMI and other variables in a cross-sectional setting. (Heuch I, Hagen K, Heuch I, Nygaard Ø, Zwart JA. The impact of body mass index on the prevalence of low back pain: the HUNT study. Spine. 2010;35:764-8.) The second paper deals with an 11-year follow-up study including 8733 men and 10 149 women, aged 30 to 69 years at start of follow-up, who did not have chronic low back pain at baseline, and 2669 men and 3899 women with low back pain at baseline. (Heuch I, Heuch I, Hagen K, Zwart JA. Body mass index as a risk factor for developing chronic low back pain: a follow-up in the Nord-Trøndelag Health Study. Spine. 2013;38:133-9.) The third paper represents a study of the comparison of anthropometric measures as body weight, BMI, waist circumference, hip circumference and waist-hip-ratio and risk of chronic low back pain. In this 11 year follow-up study of 10 059 women and 8725 men aged 30–69 years without LBP, and 3883 women and 2662 men with low back pain at baseline, associations with low back pain at end of follow-up were assessed by
--

	generalized linear modeling, with adjustment for potential confounders age, education, work status, physical activity, smoking, lipid levels and blood pressure. Positive associations with LBP at end of follow-up were all significant for body weight, BMI, waist circumference and hip circumference after similar adjustment, both in women without and with low back pain at baseline, and in men without LBP at baseline. After additional mutual adjustment for anthropometric measures, the magnitude of the association with body weight increased in women without low back pain at baseline (RR: 1.130 per standard deviation, 95% CI: 0.995–1.284) and in men (RR: 1.124, 95% CI 0.976–1.294), with other measures showing weak associations only. (Heuch I, Heuch I, Hagen K, Zwart JA. for assessing the association between body size and risk of chronic low back pain: the HUNT study. PLoS One. 2015;10:e0141268). I suspect the authors had this paper in mind when they included the reference [20] in line 15 on page 6 of 30. This might not be correct for the reference [20] on page 8 of 30 line 47, which must be the actual paper [20] included in the reference list. The authors consider the combined data set of men and women in all their analyses, although they give some separate descriptive results for each sex. Anthropometric measures do not always represent the same underlying quantities in men and women, so the reader may wonder if this is completely correct. Should not this topic be dealt with in the discussion? On page 10 of 30, line 12, the authors refer to tests of sex interaction by BMI. Is the corresponding result mentioned anywhere in the results section? This is only a minor detail, but on page 22 of 30, in Table 2, the relevant ages are only given by one number. For example, the age interval 60-64 is only represented by 63. What is the reason for this? In Table S3 the authors describe an analysis with interaction between age and BMI. Can the terms for main effects be given any meaningful interpretation in this case? It seems that BMI must be equal to zero for the main effect to be interpretable. In particular, it is not clear what kind of hypothesis the p-values apply to.
--	---

REVIEWER	Takafumi Abe Shimane University
REVIEW RETURNED	25-Jun-2020

GENERAL COMMENTS	This manuscript does not follow the author guidelines in the BMJ Open and STROBE checklist. Therefore, reviewers need a lot of effort to review this incomplete manuscript. Abstract (1)Is the study design the cross-sectional and/or prospective design? (2)Please describe the primary outcome (evaluation method and definition) accurately. I'm afraid the reader will misunderstand. Introduction (1)"its prevalence is projected to increase as population ageing continues is a considerable cause for concern." Why did the prevalence of pain decrease after 53 years old in your participants? I didn't find the reason and/or comparison of previous studies in the discussion section.
---

	(2)The pain evaluated with interviews and that evaluated with the questionnaire are compared as well. Some previous studies also evaluated pain through different methods. Please confirm how the previous studies you cited evaluated pain and revise the Introduction and Discussion. (3)Please make the figure about this study design (and analysis plan) for readers. In the Introduction and/or methods, you need to explain the reason of this study design. Why did you conduct to analyze using both cross-sectional and prospective design? Why did BMI and WC at aged 69 years assess after pain aged 68 years? STROBE recommends "Present key elements of study design early in the paper". Methods (1)It's not clear if the pain assessment method is validated in this study. How is reliability? Studies using questionnaires that have not been validated in the population of interest may be subject to measurement error, and any conclusions drawn cannot be made with total confidence. Please describe the interview method (e.g. inter-rater reliability) accurately. (2)I concern the many missing data in this study. There was no bias in the data, but were the results affected? (3)Please provide information about multicollinearity (e.g. height and BMI/WC). Is the increase in BMI caused by a decrease in height? Why did not you use "residential country/city, height loss, pain at a previous point, and/or pain history" as confounders? Please conduct the analyses using these confounders. (4)Why did not you define the statistical significance level? Table (1)Table 1 showed the chi-square and t-test as statistical strategies. However, I did not find this description in the methods. Why did you examine sex-difference in Table 1? Despite no sex-difference (36, 43, 53, and 60-64) in Table 1, you made model 1(including sex). Please add information about models 1 and 2 in the methods. (2)Please describe "age (60-64 or 63?)" in the Table1 and 2/3 accurately. Did you consider to analyze the difference of assessment points aged 60-64 years? Did you use 63 only? (2) Please describe all analyze strategies detail on the footnote in the Table2 and 3.
--	--

REVIEWER	Aleksander Galas Jagiellonian University, Faculty of Medicine, Chair of EPidemiology and Preventive Medicine, Poland
REVIEW RETURNED	04-Aug-2020

GENERAL COMMENTS	The submitted manuscript addresses an interesting issue on the change of the impact / association between BMI and back pain with age. In my opinion the manuscript may provide valuable information about this phenomenon, and the available research data gives an opportunity for in-depth look at the topic. I believe the manuscript may benefit by taking into consideration the following remarks:
---

	Major points: Abstract *-I suggest adding the purpose of the study Article summary (page 3) L8 – the descriptive data on pain is very scarce, the prospective observation should enable to present incidence and changes, so if the authors want to present this point as an added value I suggest adding more data. L15 – Authors analysed SDs of the BMI and WC, so they changed the point of analysis and interpretation. The point presented in the current form may be misleading to the readers. L24 – as the back pain was not assessed in the same way the variability in OR / or lack of the variability may be caused by the differences in the measurements, so in my opinion the second part is not justified here. Main manuscript: *-the sampling method is not presented in a content, please add. Additionally, the clarification is need whether the same individuals (sampled once at the beginning of this 'long term project') were investigated in each wave or the samples, sampled from the same birth cohort (of March 1946) were used. *-provide please, data, justifying the representativeness of the sample (including response rate), considering also the possibility of selection bias *-as mentioned by authors, there were 24 assessments made in the NSHD project, so why did the authors choose and present only 5 of them? Clarification is needed *-as I understand the analysis of the association between BMI and back pain was not within the primary purposes of the NSMD, so it would be mentioned that this is a post hoc analysis. *-Additionally, I would encourage to add a paragraph about the required sample size. *-I do not clearly understand what is the purpose to present the depiction as presented in lines 12-22. I would be better to present the exact data collection method used *-provide more details about back pain assessments (the original sounding of the questions asked ...) "all or most of the time" over a period of ??? last 10y ? Provide please the rationale for comparing recurring/severe backache with any ache or pain and additionally across different periods of time P7.L10-15 – although this is a strategy which helps in statistical analyses and the strategy of this type is typically used in the scale development or so, I would suggest strongly avoiding this here, as I do not think this is a good strategy for the measurement of the association between BMI and back pain. The BMI is a commonly used measure, with accepted norms and standardized measurement technique and the impact of the BMI as expressed in kg/m² is easily understandable and comparable. SD represents the variability level within the sample, and is clearly related to the values measured. As the SD differs across populations, and/or cohorts the associations found in this study won't be comparable with others. They do not provide information about the magnitude of the effect, which may / might be expected in populations/cohorts of interest by public health professionals. The health professional is interested in how the change in 1 BMI is associated with back pain, but not how the change by 1 SD ... ! Covariates
--	--

	*- I strongly suggest adding painkillers use, acupuncture, and / or rehabilitation, also the diagnosis of muscle-skeletal disorders should be considered. I believe this data should be available in NSHD, especially as it was mentioned participants contacted clinics. Statistical analysis *-As the authors tried to analyse whether the relationship between BMI and back pain changes over time, it is not clear to my why they did not use trajectory analysis. *-P7.L10 – provide, please, standardization formula – if there will be SDs finally presented (instead of BMIs) *- As I understand the authors used SDs of BMI – so this information should be clearly stated (but not only BMI ...) Results *-in the manuscript authors comment the presence of the association even if the results are not statistically significant. Insignificant result DOES NOT support any conclusion making! Correct please. If the p is 0.05 or more it IS NOT ‘some evidence’ Discussion I suggest creating separate paragraphs on strengths and limitations of the study. The discussion including age effects, period effects and cohort effects should be included. Possible biological mechanisms which may have stable impact on the back pain risk (and no role of others) should be discussed (if there is such effect as proposed by authors ...) Consider, please, general adjustments to the improvement suggestions mentioned above. Minor issues: P2.L.54 – the statement is not clear to me P5.L5/6 – the obesity is a target for intervention being a main risk factor for the whole group of so called dietary related diseases, so I would reword this as ‘additional benefit’ P5.L19/20 – although BMI is a general measure of overweight and obesity there is a lot of discussion whether BMI is a good measure of adiposity –so re-word the sentence. P9.L17 – what means body size here?
--	---

REVIEWER	Sarah Lay-Flurrie University of Oxford, UK
REVIEW RETURNED	10-Aug-2020

GENERAL COMMENTS	This paper examines the relationship between body mass index/ waist circumference and back pain over the life course, using data from the MRC National Survey of Health and Development. Overall, this is a well written manuscript with clearly described methods and appropriate statistical analysis. The additional analysis and consideration of interaction terms is extensive and should be commended. I have only a few minor comments:  1. P10, Line 33-34: It would be helpful to state the comparator wave as age 36 earlier here, as a reminder to the reader and to avoid confusion with the later statements in the paragraph 2. P12 lines 17-29. The comments about stronger association at age 60-64 are slightly too strong, given the number of tests conducted and the p-value for the interaction term of 0.07. The
--

	authors also appear to contradict themselves by ending the paragraph by saying that "associations at each age were fairly constant" and their arguments here could be clarified. 3. Similar to point 2 above, the first sentence of the Abstract results section is slightly too strong.
--	--

VERSION 1 – AUTHOR RESPONSE

Reviewer 1

The study investigates associations of body mass index (BMI) and waist circumference with back pain. The paper is well-written and includes data collected over a long period of time. This represents a valuable source of data compared to other studies of back pain.

Response: We thank the reviewer for their positive assessment of our paper.

*I would have preferred that the authors had included as references more papers of studies of the association between these anthropometric measures and back pain, and I have suggested the three studies published in papers mentioned below. Especially I think it is important that the authors are aware of the third paper I mention, which deals with a comparison of anthropometric measures and back pain in a follow-up setting. The first paper I recommend that the authors should be aware of included 30 102 men and 33 866 women with information on BMI and chronic low back pain. The age-groups included were 20-39 years, 40-49 years, 60-79 years, and the last age group included participants who were at least 80 years old. Relations were assessed by logistic regression of low back pain with respect to BMI and other variables in a cross-sectional setting. (Heuch I, Hagen K, Heuch I, Nygaard Ø, Zwart JA. The impact of body mass index on the prevalence of low back pain: the HUNT study. *Spine*. 2010;35:764-8.) The second paper deals with an 11-year follow-up study including 8733 men and 10 149 women, aged 30 to 69 years at start of follow-up, who did not have chronic low back pain at baseline, and 2669 men and 3899 women with low back pain at baseline. (Heuch I, Heuch I, Hagen K, Zwart JA. Body mass index as a risk factor for developing chronic low back pain: a follow-up in the Nord-Trøndelag Health Study. *Spine*. 2013;38:133-9.) The third paper represents a study of the comparison of anthropometric measures as body weight, BMI, waist circumference, hip circumference and waist-hip-ratio and risk of chronic low back pain. In this 11 year follow-up study of 10 059 women and 8725 men aged 30–69 years without LBP, and 3883 women and 2662 men with low back pain at baseline, associations with low back pain at end of follow-up were assessed by generalized linear modeling, with adjustment for potential confounders age, education, work status, physical activity, smoking, lipid levels and blood pressure. Positive associations with LBP at end of follow-up were all significant for body weight, BMI, waist circumference and hip circumference after similar adjustment, both in women without and with low back pain at baseline, and in men without LBP at baseline. After additional mutual adjustment for anthropometric measures, the magnitude of the association with body weight increased in women without low back pain at baseline (RR: 1.130 per standard deviation, 95% CI: 0.995–1.284) and in men (RR: 1.124, 95% CI 0.976–1.294), with other measures showing weak associations only. (Heuch I, Heuch I, Hagen K, Zwart JA. for assessing the association between body size and risk of chronic low back pain: the HUNT study. *PLoS One*. 2015;10:e0141268). I suspect the authors had this paper in mind when they included the reference [20] in line 15 on page 6 of 30. This might not be correct for the reference [20] on page 8 of 30 line 47, which must be the actual paper [20] included in the reference list.*

Response: We thank the reviewer for highlighting these important papers. Our work had been informed by these previous studies but in selecting which references to cite we had chosen to focus on systematic reviews. We appreciate that this was an oversight and so all three references referred to by the reviewer are now cited in our paper's introduction and the reference list has been updated accordingly (see references 13, 14 and 22).

The authors consider the combined data set of men and women in all their analyses, although they give some separate descriptive results for each sex. Anthropometric measures do not always represent the same underlying quantities in men and women, so the reader may wonder if this is

completely correct. Should not this topic be dealt with in the discussion? On page 10 of 30, line 12, the authors refer to tests of sex interaction by BMI. Is the corresponding result mentioned anywhere in the results section?

Response: We presented descriptive statistics in table 1 stratified by sex as there was evidence that the distributions of some variables varied by sex. However, the presence of sex differences in the distributions of variables does not automatically imply that there will be sex differences in associations between these variables. We have now clarified in the methods (page 9) that “Formal assessment of whether associations between BMI and back pain varied by sex were performed by including sex by BMI interaction terms in models and where no evidence of interaction was found models were sex-adjusted.” We also now refer to the fact that no evidence of interactions between sex and BMI or sex and waist circumference was found and report p-values from formal tests of this in the second paragraph of the results section (page 10) and in the footnotes to tables 2 and 3. We apologise for this omission in the previous version of our manuscript.

This is only a minor detail, but on page 22 of 30, in Table 2, the relevant ages are only given by one number. For example, the age interval 60-64 is only represented by 63. What is the reason for this?

Response: Unlike all other assessments, which were conducted within a one-year timeframe, the age range for this particular assessment was 60-64 and the mean age of assessment was 63. However, we recognise that it is confusing to switch between these two sets of values in our reporting and so for consistency we have updated the values in tables 2 and 3 to refer to 60-64

In Table S3 the authors describe an analysis with interaction between age and BMI. Can the terms for main effects be given any meaningful interpretation in this case? It seems that BMI must be equal to zero for the main effect to be interpretable. In particular, it is not clear what kind of hypothesis the p-values apply to.

Response: As BMI and waist circumference were standardised, so have means of 0 and SDs of 1 (see methods, page 7), the main effects of age presented in table S3 represent the estimates at the mean BMI/WC. We have added a footnote to table S3 to clarify this.

Reviewer 2

This manuscript does not follow the author guidelines in the BMJ Open and STROBE checklist. Therefore, reviewers need a lot of effort to review this incomplete manuscript.

Response: Our paper had been formatted according to the author guidelines for J Epidemiol Community Health from where it was transferred to BMJ Open. We assume that the BMJ Open editorial team would have alerted us if we had not adhered to important guidelines and asked us to make amendments prior to review if this had been necessary. A STROBE checklist was provided with our submission and so we are surprised by the reviewer’s assessment that this had not been followed.

Abstract

(1)Is the study design the cross-sectional and/or prospective design?

Response: As per the STROBE checklist, we indicated the study’s design in the title and abstract of our paper with inclusion of the term ‘British birth cohort’. This is in line with how the study has been described in other papers we have published in BMJ Open (see for example, Kuh et al 2019;9:e025755).

Prospectively ascertained data from this longitudinal study were examined in this paper using both cross-sectional and longitudinal analyses. To provide further clarification on this we now refer to the fact that we aimed to test both cross-sectional and longitudinal associations in the objectives section of the abstract.

(2)Please describe the primary outcome (evaluation method and definition) accurately. I'm afraid the reader will misunderstand.

Response: Our primary outcome was back pain. As requested, we have added more detail on the methods of ascertainment of this measure at the 5 ages at which it was assessed to the abstract. We

have also removed reference to our statistical models from this section to further reduce any misunderstanding.

Introduction

(1)"its prevalence is projected to increase as population ageing continues is a considerable cause for concern." Why did the prevalence of pain decrease after 53 years old in your participants? I didn't find the reason and/or comparison of previous studies in the discussion section.

Response: We believe the statement in our introduction is a fair point to make as this was hypothesised before conducting analysis of our data.

In relation to our findings, we interpret the prevalence as remaining constant after 53 years of age, with caution required in interpreting minor differences in prevalence between ages given these are prevalence estimates with associated error and there were changes in how back pain was assessed at different ages. For these reasons, we have deliberately not commented on changes in prevalence with age. However, we have now included comment on how the prevalence estimates in our study compare with previous studies in the discussion (page 13).

(2)The pain evaluated with interviews and that evaluated with the questionnaire are compared as well. Some previous studies also evaluated pain through different methods. Please confirm how the previous studies you cited evaluated pain and revise the Introduction and Discussion.

Response: As requested, we have revised the introduction (page 4) and discussion (page 13) to indicate how back pain has been assessed in other studies. We hope this will reassure readers that our methods are similar to those commonly used in other population-based studies of back pain.

(3)Please make the figure about this study design (and analysis plan) for readers. In the Introduction and/or methods, you need to explain the reason of this study design. Why did you conduct to analyze using both cross-sectional and prospective design? Why did BMI and WC at aged 69 years assess after pain aged 68 years? STROBE recommends "Present key elements of study design early in the paper".

Response: We have amended the final paragraph of the introduction to clarify our reason for conducting cross-sectional and longitudinal analyses (page 5). Other key elements of our study design are presented immediately after, on pages 5 to 10.

Data for these analyses are drawn from a large population-based study designed to capture information on a wide range of different measures of health and their risk factors across life rather than to address our specific research question. There are some unavoidable limitations resulting from this. At the latest wave of data collection, so that objective measurements of health (including height, weight and waist circumference) could be prioritised at the home visit during which nurses had only limited time available to collect data, a decision was taken to ascertain other measures (including back pain) in a postal questionnaire sent in advance. As a consequence of this, data on back pain were assessed at age 68 years and BMI and WC were assessed at age 69 years. We acknowledge that this is a limitation and now comment on this in the discussion (page 14).

Methods

(1)It's not clear if the pain assessment method is validated in this study. How is reliability? Studies using questionnaires that have not been validated in the population of interest may be subject to measurement error, and any conclusions drawn cannot be made with total confidence. Please describe the interview method (e.g. inter-rater reliability) accurately.

Response: We have clarified in the methods (page 6) that nurse interviews were structured and have provided the specific questions asked at each age as supplementary information (supplementary methods). Although our methods of ascertainment of back pain are similar to those commonly used in other population-based studies and our prevalence estimates are comparable (page 13) suggesting good face validity we now acknowledge in the discussion that it is a limitation of our study that the validity of the back pain assessments used has not been formally evaluated (page 14).

(2)I concern the many missing data in this study. There was no bias in the data, but were the results affected?

Response: Table S1 which details the number of people with missing data was purposefully provided to ensure transparency. As the repeated analysis approach used includes any individual with at least one outcome measure and is valid under a missing at random assumption we have maximised the number of participants included in the analysis which will help to minimise bias. We also now acknowledge that those people who were lost to follow-up before age 36 and/or did not provide any data on back pain and body size could not be included in analyses and that this is a limitation (page 14).

(3) Please provide information about multicollinearity (e.g. height and BMI/WC). Is the increase in BMI caused by a decrease in height? Why did not you use "residential country/city, height loss, pain at a previous point, and/or pain history" as confounders? Please conduct the analyses using these confounders.

Response: The purpose of deriving BMI is to create a measure of weight independent of height, we therefore have no concerns about multicollinearity in models including both BMI and height as by design these measures are not correlated. Correlations between WC and height are also not sufficiently high for us to have concerns about including WC and height in the same models and when we did this there was no evidence of multicollinearity.

We are confident that the increase in BMI observed with increasing age is largely attributable to increases in weight (adiposity) rather than to the very small observed decreases in mean height, especially as these small changes in height may in part be attributable to measurement error and changes in the sample size over time.

As outlined on page 8, we chose to adjust for covariates selected *a priori* which represent different domains of the biopsychosocial model of pain including variables that have previously been identified as key risk factors for back pain and so could potentially confound associations. On this basis we cannot justify inclusion of residential country/city. We do include adjustment for height but cannot also include height loss as such a measure would contain considerable error as it will be largely attributable to random variation between measures. Adjustment for history of pain or a previous measure of pain would be addressing a different research question as we would then be investigating the association between BMI and current pain conditional on previous pain. Previous pain is thus not a confounding variable.

(4) Why did not you define the statistical significance level?

Response: In our interpretation and reporting of results we were guided by a number of key papers which have clearly highlighted the need to move away from reporting results as 'significant' or 'non-significant' based on an arbitrary threshold (usually $P < 0.05$), see for example, Sterne and Davey Smith 'Sifting the evidence – what's wrong with significance tests?' *BMJ* 2001;322:226-31; Amrhein, Greenland and McShane (plus 800 signatories) 'Retire statistical significance' *Nature* 2019;567:305-307; Wasserstein, Schirm and Lazar 'Moving to a world beyond "p<0.05"' *The American Statistician* 2019;73, suppl1:1-19; Watt 'Reflections on modern methods: Statistics education beyond 'significance'' *Int J Epidemiol* 2020 <https://doi.org/10.1093/ije/dyaa080>

Following this guidance we chose to focus our reporting and interpretation on effect estimates, 95% confidence intervals and exact p-values.

Table

(1) Table 1 showed the chi-square and t-test as statistical strategies. However, I did not find this description in the methods. Why did you examine sex-difference in Table 1? Despite no sex-difference (36, 43, 53, and 60-64) in Table 1, you made model 1 (including sex). Please add information about models 1 and 2 in the methods.

Response: We apologise for this omission. The fact that we first examined descriptive statistics for each variable and formally tested sex differences is now reported on page 9. We presented descriptive statistics in table 1 stratified by sex as there was evidence that the distributions of some variables varied by sex. We subsequently included sex in our models because it had been selected *a priori* as a potential confounder.

We have revised the description of our statistical analyses (page 9) to clarify which models are models 1 and 2 as shown in the results tables.

(2) Please describe "age (60-64 or 63?)" in the Table 1 and 2/3 accurately. Did you consider to analyze the difference of assessment points aged 60-64 years? Did you use 63 only?

Response: As noted in response to reviewer 1, unlike all other assessments, which were conducted within a one-year timeframe, the age range for this particular assessment was 60-64 and the mean age of assessment was 63. However, we recognise that it is confusing to switch between these two sets of values in our reporting and so for consistency we have updated the values in tables 2 and 3 to refer to 60-64.

(2) Please describe all analyze strategies detail on the footnote in the Table 2 and 3.

Response: We have added additional details to the footnotes of Tables 2 and 3.

Reviewer 3

The submitted manuscript addresses an interesting issue on the change of the impact / association between BMI and back pain with age. In my opinion the manuscript may provide valuable information about this phenomenon, and the available research data gives an opportunity for in-depth look at the topic.

Response: We thank the reviewer for their positive assessment of our paper.

I believe the manuscript may benefit by taking into consideration the following remarks:

Major points:

Abstract

**-I suggest adding the purpose of the study*

Response: In adhering to the BMJ Open guidelines on abstract headings, the purpose of our study is outlined under the heading 'Objectives'.

Article summary (page 3)

L8 – the descriptive data on pain is very scarce, the prospective observation should enable to present incidence and changes, so if the authors want to present this point as an added value I suggest adding more data.

Response: Thank you for this suggestion. While a clear strength of our study is the availability of data on back pain ascertained over 32 years of follow-up, monitoring of pain over this period was at 5 specific ages (36, 43, 53, 60-64 and 68 years). As there were gaps of up to 10 years between assessments and we did not ascertain information on the timing of the onset of back pain unfortunately we are unable to reliably estimate incidence of back pain. In addition, while we report the prevalence of back pain at each age of assessment over this 32 year time period, we have purposefully not reported on changes in prevalence over time given changes between ages in the methods of assessment of back pain. These limitations of our study are acknowledged (see 3rd bullet point of the article summary and discussion, page 14).

L15 – Authors analysed SDs of the BMI and WC, so they changed the point of analysis and interpretation. The point presented in the current form may be misleading to the readers.

Response: As outlined in the methods (page 7), BMI and WC were sex-standardised to facilitate comparisons of effect sizes across age and sex. They also allow for a fairer comparison of the effects for BMI with the effects for WC and overcome the limitation that there is no standard unit of analysis for waist circumference.

This involves a simple rescaling of BMI and WC so that the means are 0 and SDs are 1. Findings from these models are equivalent to those from models in which BMI and WC are included in their raw units (i.e. kg/m² and cm, respectively). This approach, which is commonly employed, does not

change the point of analysis or interpretation. To provide reassurance of this we have added results from models in which BMI and WC are modelled in raw units (i.e. kg/m² and cm) as supplementary information (see Tables S5 and S6).

L24 – as the back pain was not assessed in the same way the variability in OR / or lack of the variability may be caused by the differences in the measurements, so in my opinion the second part is not justified here.

Response: We have deleted the second part of the third bullet point of the article summary in response to this comment.

Main manuscript:

**-the sampling method is not presented in a content, please add. Additionally, the clarification is need whether the same individuals (sampled once at the beginning of this 'long term project') were investigated in each wave or the samples, sampled from the same birth cohort (of March 1946) were used.*

Response: As outlined in the opening sentence of the subjects and methods section “The NSHD is a socially stratified sample of 5362 single, legitimate births that occurred in England, Wales and Scotland in one week of March 1946.” We have now clarified in the following sentence that this study has followed up the same participants across life and cite key papers which provide further details of the study design including the target sample and sampling strategy (pages 5-6).

**-provide please, data, justifying the representativeness of the sample (including response rate), considering also the possibility of selection bias*

Response: We now refer to the fact that participation rates have remained relatively high in our study across life (page 6) and 3 papers are cited (refs 23-25) which provide more details on the representativeness of our sample. We also now discuss the potential for bias due to loss to follow-up in the discussion (page 14).

**-as mentioned by authors, there were 24 assessments made in the NSHD project, so why did the authors choose and present only 5 of them? Clarification is needed*

Response: Back pain was ascertained during 5 of the 24 assessments of NSHD participants. We could not include any more waves of data as back pain was not ascertained during the other 19 main assessments. This is now clarified in the opening sentence on back pain assessment (page 6).

**-as I understand the analysis of the association between BMI and back pain was not within the primary purposes of the NSMD, so it would be mentioned that this is a post hoc analysis.*

Response: Data for these analyses are drawn from a large population-based study designed to capture information on a wide range of different measures of health and their risk factors across life rather than to address this specific research question. This is very common in epidemiology and researchers are actively encouraged to address interesting and novel research questions using these incredibly extensive existing data resources to maximise their value. We now acknowledge the fact that our analyses are post hoc in the discussion (page 14).

**-Additionally, I would encourage to add a paragraph about the required sample size.*

Response: We have never been asked to provide this information in our prior BMJ publications using the same study. Given our sample size is fixed because of its birth cohort study design and, associations were found suggesting we did have sufficient statistical power we do not feel that this would be appropriate or informative.

**-I do not clearly understand what is the purpose to present the depiction as presented in lines 12-22. I would be better to present the exact data collection method used*

Response: We are unsure what the reviewer is referring to.

**-provide more details about back pain assessments (the original sounding of the questions asked ...) "all or most of the time" over a period of ??? last 10y ? Provide please the rationale for comparing recurring/severe backache with any ache or pain and additionally across different periods of time*

Response: We have added copies of the questions used to assess back pain at each of the 5 ages to Supplementary methods.

As noted above, data for these analyses are drawn from a large population-based study designed to capture information on a wide range of different measures of health and their risk factors across life rather than to address this specific research question. Our decision to compare back pain assessed in slightly different ways at different ages and measured across different periods of time was therefore a pragmatic one. We are confident that the analytical approach we took, though post-hoc (as now acknowledged in the discussion), makes the best possible use of the data available and despite acknowledged limitations adds important new insights to the literature.

P7.L10-15 – although this is a strategy which helps in statistical analyses and the strategy of this type is typically used in the scale development or so, I would suggest strongly avoiding this here, as I do not think this is a good strategy for the measurement of the association between BMI and back pain. The BMI is a commonly used measure, with accepted norms and standardized measurement technique and the impact of the BMI as expressed in kg/m² is easily understandable and comparable. SD represents the variability level within the sample, and is clearly related to the values measured. As the SD differs across populations, and/or cohorts the associations found in this study won't be comparable with others. They do not provide information about the magnitude of the effect, which may / might be expected in populations/cohorts of interest by public health professionals. The health professional is interested in how the change in 1 BMI is associated with back pain, but not how the change by 1 SD ... !

Response: As outlined in the methods (page 7), BMI and WC were sex-standardised to facilitate comparisons of effect sizes across age and sex. This method is commonly employed and involves a simple rescaling of BMI and WC so that the means are 0 and SDs are 1. After careful consideration we have decided to leave these analyses in as they do allow for a fairer comparison of effect sizes across age (uninfluenced by the widening distribution of BMI with age) and also for a comparison between the effect sizes for BMI and WC within our study. However, to address the reviewer's concern and to make it clearer how results from our study relate to those in other studies with different distributions of BMI we have added supplementary tables (Tables S5 and S6) showing the results from our main models rerun with BMI modelled as kg/m² and WC modelled as cm. As can be seen from these tables, our findings and main conclusions remain the same.

Covariates

**- I strongly suggest adding painkillers use, acupuncture, and / or rehabilitation, also the diagnosis of muscle-skeletal disorders should be considered. I believe this data should be available in NSHD, especially as it was mentioned participants contacted clinics.*

Response: As noted in a response to reviewer 2 above, we chose to adjust for covariates selected *a priori* which represent different domains of the biopsychosocial model of pain including variables that have previously been identified as key risk factors for back pain. The variables suggested by the reviewer were not identified *a priori* and as they are likely to be on the causal pathway would not meet the definition of a confounder.

Statistical analysis

**-As the authors tried to analyse whether the relationship between BMI and back pain changes over time, it is not clear to my why they did not use trajectory analysis.*

Response: We gave very careful consideration to the most appropriate way to model the available data in order to best contribute to the existing literature on the association between adiposity and back pain. As the measures of back pain in NSHD were ascertained at only 5 time points over a period of 30 years, it is known that back pain comes and goes over time, and back pain was ascertained in slightly different ways at different ages, we felt that modelling back pain as a trajectory with age for the purposes of addressing this research question was inappropriate.

**-P7.L10 – provide, please, standardization formula – if there will be SDs finally presented (instead of BMIs)*

Response: We have added the formula used to standardise BMI and WC to the methods, page 7.

**- As I understand the authors used SDs of BMI – so this information should be clearly stated (but not only BMI ...)*

Response: We have ensured that we clearly and consistently refer to the use of SDs in results and in the tables.

Results

**-in the manuscript authors comment the presence of the association even if the results are not statistically significant. Insignificant result DOES NOT support any conclusion making! Correct please. If the p is 0.05 or more it IS NOT 'some evidence'*

Response: As noted in our response to reviewer 2, in our interpretation and reporting of results we were guided by a number of key papers which have clearly highlighted the need to move away from reporting results as 'significant' or 'non-significant' based on an arbitrary threshold (usually $P < 0.05$), see for example, Sterne and Davey Smith 'Sifting the evidence – what's wrong with significance tests?' *BMJ* 2001;322:226-31; Amrhein, Greenland and McShane (plus 800 signatories) 'Retire statistical significance' *Nature* 2019;567:305-307; Wasserstein, Schirm and Lazar 'Moving to a world beyond "p<0.05"' *The American Statistician* 2019;73, suppl1:1-19; Watt 'Reflections on modern methods: Statistics education beyond 'significance'' *Int J Epidemiol* 2020
<https://doi.org/10.1093/ije/dyaa080>

Following this guidance we chose to focus our reporting and interpretation on effect estimates, 95% confidence intervals and precise p-values.

Discussion

I suggest creating separate paragraphs on strengths and limitations of the study.

Response: We have created separate paragraphs on strengths and limitations of the study in the discussion (pages 13-14) and also highlight both strengths and limitations of our study in the article summary.

The discussion including age effects, period effects and cohort effects should be included.

Response: Given the cohort is all of exactly the same age, we can be certain we are investigating age effects. However, the results observed may be specific to this particular cohort and we discuss this on page 14.

Possible biological mechanisms which may have stable impact on the back pain risk (and no role of others) should be discussed (if there is such effect as proposed by authors ...) Consider, please, general adjustments to the improvement suggestions mentioned above.

Response: In our introduction we suggested that there are reasons why associations between adiposity and back pain could change with age. However, as our findings provided no clear evidence that associations did change markedly with age we chose not to speculate on factors that may explain changes with age further in our discussion.

Minor issues:

P2.L.54 – the statement is not clear to me

Response: We have revised the final sentence of the abstract and hope it is now clear.

P5.L5/6 – the obesity is a target for intervention being a main risk factor for the whole group of so called dietary related diseases, so I would reword this as 'additional benefit'

Response: We have amended the sentence to make it clear that targeting obesity is also important for the prevention of many other chronic conditions (page 5).

P5.L19/20 – although BMI is a general measure of overweight and obesity there is a lot of discussion whether BMI is a good measure of adiposity –so re-word the sentence.

Response: Thank you for reminding us of this discussion and the need to be cautious in how we refer to BMI. To address this point we have removed the phrase ‘as a general marker of total adiposity’ from the end of this sentence.

P9.L17 – what means body size here?

Response: Thanks for spotting this error – this should have read BMI not body size and has now been corrected.

Reviewer 4:

This paper examines the relationship between body mass index/ waist circumference and back pain over the life course, using data from the MRC National Survey of Health and Development. Overall, this is a well written manuscript with clearly described methods and appropriate statistical analysis. The additional analysis and consideration of interaction terms is extensive and should be commended.

Response: We thank the reviewer for their positive assessment of our paper.

I have only a few minor comments:

1. P10, Line 33-34: It would be helpful to state the comparator wave as age 36 earlier here, as a reminder to the reader and to avoid confusion with the later statements in the paragraph

Response: Thank you for this suggestion, we have reworded this sentence to make it clear at the very start that age 36 is the comparator wave and hope that this now avoids confusion with the later statements in the paragraph (page 11).

2. P12 lines 17-29. The comments about stronger association at age 60-64 are slightly too strong, given the number of tests conducted and the p-value for the interaction term of 0.07. The authors also appear to contradict themselves by ending the paragraph by saying that "associations at each age were fairly constant" and their arguments here could be clarified.

Response: Thank you for this comment. We agree with the reviewer and so have removed the sentences on there being a stronger association at age 60-64 from the discussion.

3. Similar to point 2 above, the first sentence of the Abstract results section is slightly too strong.

Response: We have amended the first sentence of the abstract results section by removing reference to the stronger association at age 60-64 in order to tone down the statement.

VERSION 2 – REVIEW

REVIEWER	Ingrid Heuch Department of Research, Innovation and Education, Division of Clinical Neuroscience, Oslo University Hospital, Oslo, Norway
REVIEW RETURNED	02-Oct-2020
GENERAL COMMENTS	Thank you for the responses. I have no more comments. This is an informative study and represents important work in the field of anthropometric measures and back pain.
REVIEWER	S Lay-Flurrie

	University of Oxford
REVIEW RETURNED	01-Oct-2020

GENERAL COMMENTS	All of my comments have been addressed. This remains a good paper.
--